# Near Real-Time Change Detection System Using Sentinel-2 and Machine Learning: A Test for Mexican and Colombian Forests

Ana María Pacheco-Pascagaza [1,2,3,*,†], Yaqing Gou [1,2,4,†], Valentin Louis [1,2], John F. Roberts [1,2], Pedro Rodríguez-Veiga [1,2], Polyanna da Conceição Bispo [1,2,3], Fernando D. B. Espírito-Santo [1], Ciaran Robb [5], Caroline Upton [1], Gustavo Galindo [6], Edersson Cabrera [6], Indira Paola Pachón Cendales [6], Miguel Angel Castillo Santiago [7], Oswaldo Carrillo Negrete [8], Carmen Meneses [8], Marco Iñiguez [8] and Heiko Balzter [1,2]

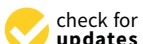



1 Centre for Landscape and Climate Research (CLCR), Space Park Leicester (SPL), School of Geography, Geology and Environment, University of Leicester, 92 Corporation Rd, Leicester LE4 5SP, UK; yaqing.gou@wur.nl (Y.G.); valentin.louis@leicester.ac.uk (V.L.); jfr10@leicester.ac.uk (J.F.R.); prv4@leicester.ac.uk (P.R.-V.); fdbes1@leicester.ac.uk (F.D.B.E.-S.); cu5@leicester.ac.uk (C.U.); hb91@leicester.ac.uk (H.B.)
2 National Centre for Earth Observation, University of Leicester, Space Park Leicester, Corporation Road, Leicester LE4 5SP, UK
3 Department of Geography, School of Environment, Education and Development, University of Manchester, Manchester M13 9PL, UK; polyanna.bispo@manchester.ac.uk
4 Laboratory of Geo-Information Science and Remote Sensing, Wageningen University & Research, Rijksweg 5, 6705 Wageningen, The Netherlands
5 UK Centre for Ecology and Hydrology, Environment Centre Wales, Deiniol Road Bangor, Gwynedd LL57 2UW, UK; ciarob@ceh.ac.uk
6 Instituto de Hidrología, Meteorología y Estudios Ambientales (IDEAM), Calle 25 D, Bogotá 110911, Colombia; ggalindo@ideam.gov.co (G.G.); ecabreram@ideam.gov.co (E.C.); ipachon@ideam.gov.co (I.P.P.C.)
7 El Colegio de la Frontera Sur (ECOSUR), Carretera Panamericana y Periférico sur s/n, San Cristóbal de las Casas 29290, Mexico; mcastill@ecosur.mx
8 Comisión Nacional Forestal (CONAFOR), Periférico Poniente 5360 San Juan de Ocotán, Jalisco 45019, Mexico; ocarrillo@colpos.mx (O.C.N.); cmeneses.ute@conafor.gob.mx (C.M.); marcoiniguez.ute@conafor.gob.mx (M.I.)
* Correspondence: ana.pachecopascagaza@manchester.ac.uk
† These authors contributed equally to this work.

**Abstract:** The commitment by over 100 governments covering over 90% of the world's forests at the COP26 in Glasgow to end deforestation by 2030 requires more effective forest monitoring systems. The near real-time (NRT) change detection of forest cover loss enables forest landowners, government agencies and local communities to monitor natural and anthropogenic disturbances in a much timelier fashion than the thematic maps that are released every year. NRT deforestation alerts enable the establishment of more up-to-date forest inventories and rapid responses to unlicensed logging. The Copernicus Sentinel-2 satellites provide operational Earth observation (EO) data from multi-spectral optical/near-infrared wavelengths every five days at a global scale and at 10 m resolution. The amount of acquired data requires cloud computing or high-performance computing for ongoing monitoring systems and an automated system for processing, analyzing and delivering the information promptly. Here, we present a Sentinel-2-based NRT change detection system, assess its performance over two study sites, Manantlán in Mexico and Cartagena del Chairá in Colombia, and evaluate the forest changes that occurred in 2018. An independent validation with very high-resolution PlanetScope (~3 m) and RapidEye (~5 m) data suggests that the proposed NRT change detection system can accurately detect forest cover loss (> 87%), other vegetation loss (> 76%) and other vegetation gain (> 71%). Furthermore, the proposed NRT change detection system is designed to be attuned using in situ data. Therefore, it is scalable to larger regions, entire countries and even continents.

**Keywords:** near real-time; vegetation change detection; machine learning; deforestation; tropical forests

## 1. Introduction

Tropical forests stabilize the world's climate and protect biodiversity [1,2]. In the context of the UN Framework Convention on Climate Change and the Paris Agreement, the initiative on "Reducing Emissions from Deforestation and forest Degradation" (REDD+) aims to protect carbon stores, biodiversity and other ecosystem services [3,4]. One of the main requirements for the implementation of REDD+ is a robust and objective measurement, reporting and verification (MRV) system that is based on satellite technology [5]. Humans have changed the natural environment on a global scale to an unprecedented degree. The global changes in atmospheric chemistry and climate have led to a new geological era called the Anthropocene [6]. Land use change is the driver with the largest global impact on forest cover since three quarters of these lands have changed, thereby diminishing the productivity of 23% of global land surface [4,7,8].

Remote sensing has provided new tools and opportunities for monitoring land cover changes in recent decades. Research has focused on the generation of tree cover maps [9,10], forest aboveground biomass maps [11,12] and land cover change maps [13], as well as improving algorithms to increase their accuracy [10,13–15]. Many of these products are produced annually or with a fixed delivery schedule, which limits their usability for enabling timely interventions when unlicensed forest cover loss is detected. Additionally, current monitoring systems do not operate at spatial scales that are sufficient to detect smaller areas of deforestation [5,16]. Examples of such global products are the tree cover change maps from Global Forest Watch using Landsat imagery and the forest/non-forest maps produced from ALOS imagery by JAXA [9,10,16,17]. However, some regional studies have used near real-time (NRT) applications that help authorities to respond quickly to events (i.e., deforestation, degradation or extreme environmental events), enabling government agencies to manage the land more effectively and supporting local decision-making with timely information [16,18–21]. NRT change alerts have been used for other purposes than in forestry, such as flood mapping [22], active wildfire monitoring, natural disasters and food security [18,23,24]. Generally, they are using imagery of medium (30 m, Landsat) to coarse spatial resolution (1 km, MODIS). Most of them rely on spectral signatures to derive indices (i.e., vegetation, fires, etc.) that allow the observation of the changes being monitored [16].

The provision of freely available and timely satellite remote sensing data has grown exponentially in recent years. With the launch of the Copernicus Sentinel satellites by the European Space Agency (ESA), there is much greater potential for monitoring land cover change in NRT at high spatial resolution. Sentinel-2 provides multi-spectral (MS) imagery at a finer spatial and temporal resolution compared to Landsat 8 and 9. The 10 m spatial resolution enables Sentinel-2 to map even small plantations, clearances and wind throw in complex forest landscapes [15]. With a 5-day return period, Sentinel-2 can be used for NRT monitoring and has a higher chance of acquiring a cloud-free image than Landsat. Various papers have demonstrated the capability of Sentinel-2 data in detecting NRT forest change by combining different sensors [25–27], and particularly by using freely available cloud processing platforms such as Google Earth Engine [28].

Alongside the satellite data revolution, change detection methods have advanced dramatically in the fields of remote sensing, image processing and computer vision [29]. General review papers include critical syntheses of techniques that include image algebra, trend analysis, moving window calculations, dimensionality reduction, post-classification comparison and direct pixel classification, which may be used in isolation or in combination [30]. Many factors will affect the choice of the most appropriate method, such as the aims of the change detection, the temporal frequency of the observations and the

level of existing knowledge about the change. Image differencing is the simplest change detection technique since it only involves the subtraction of one image from another [31,32]. Moving window methods provide a localized alternative to image algebra, although they require post-operation analysis to provide a change product. Additionally, moving windows require the identification of a scale that is coarser than the spatial resolution of the image, which can reduce the detectability of specific small-scale features in the imagery [33]. Dimensionality reduction techniques, such as principal components analysis (PCA), produce uncorrelated Eigen images, within which a prominent representation of the change may occur [34,35]. Further processing via the use of thresholds or statistical learning is required to extract thematic change from the Eigen values. Techniques such as multi-variate alteration detection [36] use canonical correlation analysis to produce a similar output to PCA in the sense that a multi-band image is produced where any patterns of change are prominent, but they aim to eliminate some of the short comings of PCA, such as noise and linear scaling sensitivities. As with the previously mentioned techniques, further analysis, such as image thresholding or statistical learning, is required to map the thematic change classes [34–36]. One advantage of the methods outlined so far is that they provide a quantitative representation of change. However, they require either manual or automated post-processing to map the change.

Trend analysis techniques aim to analyze the variation of pixel values over time, such that a change in values between images beyond the bounds of the expected variation constitutes a change. Notable examples of such studies include those using the "Breaks for additive seasonal and trend" (Bfast) algorithm [37–41] and the continuous change detection and classification of land cover algorithm. Post-classification comparison (PCC) is another widely used change detection method that produces thematic maps of change. It is generated from statistical learning techniques and applied manually or by a set of rules [42]. The direct classification of change through a chronologically stacked pair of images reduces error propagation and has the advantage of directly classifying the signatures of change in the data. One of the most notable recent publications that utilizes this technique is Global Forest Watch [9], where a time series of global forest cover change is being produced from the year 2000 onwards. It employs a direct pixel classification of change, followed by trend analysis to identify the time when the changes occurred. Nonetheless, an NRT system requires an immediate detection of change rather than an annual retrospective approach. This requires the direct classification of a chronologically stacked image pair.

In this study, we introduce a Sentinel-2-based near real-time (NRT) forest monitoring system that is based on an open-source Python library called Python for Earth Observation (PYEO v3.6) [43], which automates data downloading, processing, analysis, validation and change alert reporting and won the Copernicus Masters' Award (Sustainable Living) in 2017. The change detection was conducted using a refined machine learning algorithm on a chronologically stacked pair of images. Reflectance bands of two image acquisitions were stacked into a bi-temporal image stack and the classifier was trained using the temporal and spectral information. To test the accuracy and scalability of the system, forest change detection was conducted during the 12 months from December 2017 to December 2018 in the Manantlán area of Mexico and Cartagena del Chairá in Colombia. The results were validated independently using PlanetScope data.

## 2. Materials and Methods

### 2.1. Near Real-Time (NRT) Change Detection System

The new NRT change detection system was designed to automate Sentinel-2 data downloading and processing, model training, image classification, the creation of change alerts and validation procedures. The Sentinel-2 satellite constellation consists of two satellites with a combined image acquisition frequency of 5 days with 10 m, 20 m and 60 m spatial resolutions and 13 spectral bands, from the visible spectrum to short-wave infrared [44].

This NRT change detection system was able to ingest all available Sentinel-2 bands. However, we only used the highest-resolution (10 m) bands to provide the maximum capacity to detect small-scale changes. The system was designed to detect changes in forest, as well as other vegetation, covering the nine classes as described in Table 1. Water bodies were masked out in the analysis.

**Table 1.** The definitions of change classes considered in the NRT change detection system.

| Classes | Definition |
|---|---|
| Stable Forest | Forest that remains forest between two observations. |
| Forest > Other Vegetation | Forest that changes to other vegetation. This class is difficult to detect during a short period of analysis. It may happen when deforested areas are in a recovery process but are still not within the definition of forest, or areas that replaced by other type of vegetation not identified as forest (i.e., bracken, agriculture, palm plantations). |
| Forest > Non-vegetation | Forest that changes to non-vegetated areas, i.e., deforestation. |
| Stable Other Vegetation | Areas that are covered by vegetation other than forest on both observation dates. |
| Other Vegetation > Forest | Other vegetation that changes into forest, i.e., areas of tree planting, afforestation or reforestation. |
| Other Vegetation > Non-vegetation | Other vegetation that changes into non-vegetated areas, i.e., change in crops, seasonal shrubs. |
| Stable Non-vegetation | Areas that are not covered by vegetation in either observation, i.e., bare soils, urban areas. |
| Non-vegetation > Forest | Non-vegetated areas that change into forest, i.e., areas of tree planting, afforestation or reforestation. This class might occur in a longer time series (i.e., > 10 years) [45,46]. |
| Non-vegetation > Other Vegetation | Non-vegetated areas that change into other vegetation, i.e., crop growth, bracken growth, succession. |

Prior to the NRT change detection analysis, a baseline forest cover map was generated. The baseline map consisted of a forest/non-forest mask, resulting from applying the trained model to the identification of these two classes over a year before the monitoring period. If a pixel was classified as forest for most image acquisitions in the given year and was not deforested within the last 3 months of the period, that pixel was identified as "forest". For the baseline, we did not consider the class "other vegetation", so as to avoid false positives due to seasonality. Therefore, areas identified as other vegetation were classified as non-vegetation on the baseline map. Once the baseline map was defined, the NRT change detection system followed the steps shown in Figure 1.

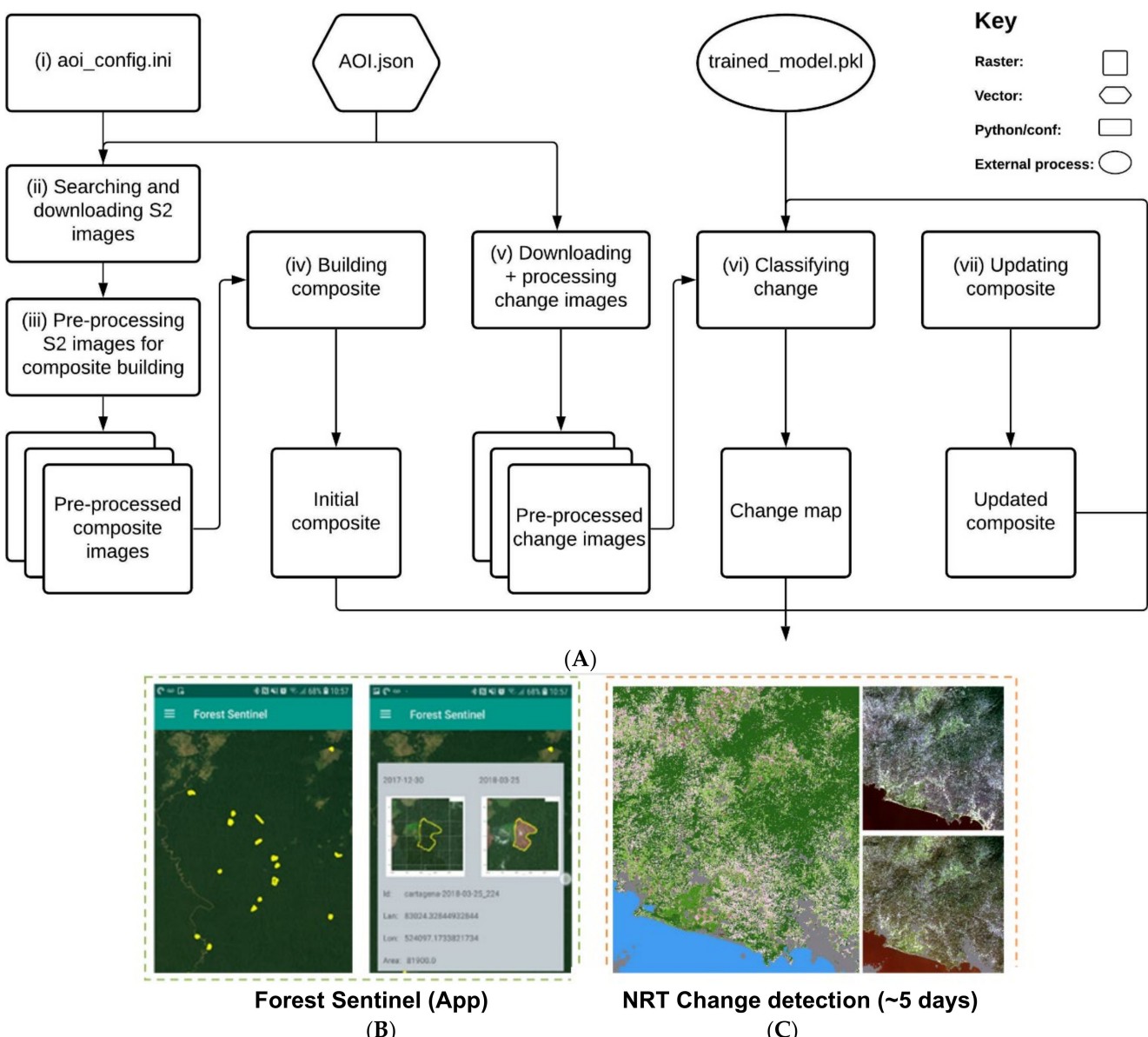

**Figure 1.** (**A**) A conceptual diagram of the proposed near real-time processing chain and outputs. S2 refers to the Sentinel-2 images. AOI refers to the area of interest or area of studio. The aoi_config.ini is the file containing all parameters for processing the Sentinel-2 image. The trained_model.pkl refers to any pre-trained machine-learning model. (**B**) An example of a deforestation alert in the Forest Sentinel App. (**C**) A quick look at the classification of change.

(i)   A configuration file was generated to write all parameters required for the processing chain, such as login information for the Copernicus hub account, path to the Sen2cor installation for cloud masking, input and output paths for the processing, etc. Further details can be found at [43].

(ii)  The system monitored the ESA Copernicus Open Data Hub to find any new Sentinel-2 data acquisitions over the area of interest on a regular basis.

(iii) Sentinel-2 imagery was acquired by an optical sensor, so the data were affected by atmospheric effects (i.e., clouds, cloud shadows, aerosols) and topographic effects (i.e., illumination angle, shadows from terrain). Masking out the pixels affected by noise and clouds was the key to reducing the omission errors in the change detection, especially for pan-tropical regions with frequent cloud cover and steep terrain. Therefore, we used a robust cloud and cloud shadow masking algorithm

by combining the cloud mask from Sen2Cor (version 2.5.5) [47], to process the top of atmosphere (TOA) L1C image to the bottom of atmosphere (BOA) reflectance at level L2A [48], with the cloud mask generated by F-mask [49] and buffered by a user-defined pixel number. Temporal gaps in the time series corresponded to areas where the cloud cover was $\geq$80%. Therefore, those images were not processed due to the likelihood of generating false positives in the change detection.

(iv) A cloud-free reference composite was generated using all available cloud-masked images from the composite-building time window for the selected area. (v) As soon as a new Sentinel-2 image was acquired and added to the Copernicus Open Access Hub, it was downloaded and processed using a pre-trained machine learning model to detect vegetation change compared to the reference image composite. The NRT element operated on the same principles as the time series data, whereby a chronologically ordered image stack, consisting of a cloud-free reference image and a newly acquired image, was processed for change detection. (vi) The last available cloud-free pixel before the current acquisition data was then used to detect change against the baseline composite. A binary mask was updated after every new image was ingested into the system, so if a pixel was identified as change, it would be eliminated from the next iteration of change detection. (vii) If visual interpretation was required for validating the resultant change maps, a set of sampling points was automatically generated in shapefile format. When a forest cover change was detected, it was archived and then disseminated in a user-friendly interactive format to an authorized server and/or registered mobile devices for further action.

In addition, this NRT system was designed to be applied to monitor local landscapes. Therefore, as vegetation types changes, it was necessary to select new AOIs representing these new vegetation types (i.e., rain forests vs. dry forests in the class "Forest" or bracken vs. crops in "Other Vegetation"). It was noticed in trials that collecting more training areas could benefit the performance and accuracy of the model. New models are generally trained over different regions to ensure that the model is adapted to local vegetation types and change patterns. If a new AOI has a distinct forest type or change pattern compared to existing AOIs, then the model will need to be updated. Moreover, we did not update the model for every new Sentinel-2 image, but rather regularly on an annual basis to achieve a better performance as we had more samples of different vegetation types represented in the same class (i.e., bracken vs. crops in "Other Vegetation"). Generally, keeping the model up to date was of importance since, for example, changes to data formats and the re-calibration of the satellite signal needed to be monitored and the models updated/changed accordingly.

The outputs consisted of: (i) a thematic map of stable and change classes; (ii) probability maps; (iii) a KMZ file with the forest cover change areas, with mean probabilities across all pixels in the contiguous polygon and their geometric properties; and (iv) quick look maps of the imagery used to detect the change for visual verification purposes (Figure 1C). The KMZ file could be displayed on any computer or mobile device and was based on Google Earth/Google Maps or integrated with axillary datasets to refine the detection results. The geospatial processing element used the PYEO v3.6 library [43]. The geospatial data outputs formed part of an integrated system that leverages mobile technology named Forest Sentinel, which won the Copernicus Masters' Sustainable Living Challenge Award in 2017 (Figure 1B). The resulting forest cover change map and alerts were sent to a server and disseminated to computers or mobile devices by registered users who could then verify the alerts in the field, allowing users to add photos, voice recordings and survey information to the data.

### 2.2. NRT Change Detection System Performance Evaluation

We evaluated the performance and accuracy of our new NRT change detection system (Section 2.1) in two study areas in the tropics: Manantlán in Mexico and Cartagena del

Chairá in Colombia (see Figure 2). The change detection was carried out with data acquired over one year, from December 2017 to December 2018.

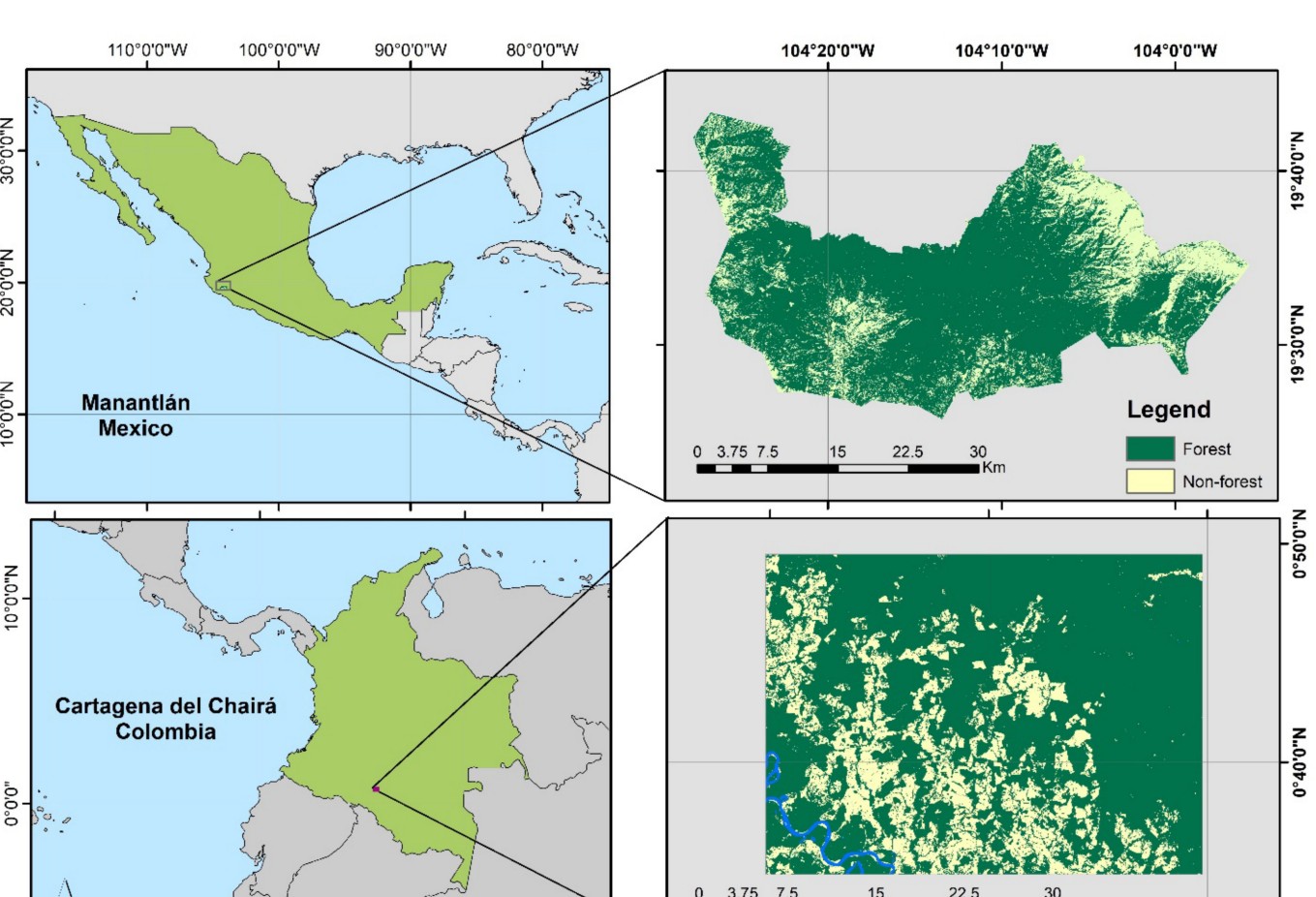

**Figure 2.** The window areas of Manantlán (Mexico) and Cartagena del Chairá (Colombia).

### 2.2.1. Study Regions

Manantlán is located in the western region of north-central Mexico, spanning the states of Jalisco and Colima. Within the region is the biosphere reserve Sierra de Manantlán, which is one of the most important natural mountain reserves in Mexico due to its biological richness, extent, environmental services and water resources. The forest vegetation is a mixture of temperate and cloud forests. Shrubs, grassland and agricultural areas are also present [50]. The forests of Manantlán have been subject to deforestation and forest fragmentation due to extensive cattle ranching and the timber extraction of oak and pine [46,51]. In addition, illegal logging and gold mining have affected the area significantly in the last decade [52,53]. Forest fires are a frequent phenomenon in Manantlán, and about 18,000 ha were affected in 2017. Most were surface fires that mainly affected leaf litter and shrub vegetation. Consequently, they did not contribute significantly to deforestation. However, in cases where the fires have affected the canopy, they could be identified as deforestation [54,55].

Cartagena del Chairá is located in the Colombian Amazon, in the Caquetá department. This region is covered by tropical moist broadleaf forest [56] and is a transition between the Orinoquia savannahs and Amazonia. The region is known for having the highest biological diversity in the country, holding a great variety of flora and fauna [57,58]. Land cover is characterized by undisturbed and natural forests that are subject to some human intervention, shrubs and grasslands, which are mainly for cattle ranching [59]. The Caquetá

Department historically has the largest deforestation rate in Colombia. Cartagena del Chairá presented the largest deforestation rate in 2015 (23,812 ha deforested), which was equivalent to 19% of the national deforestation [60,61], and the second largest in 2017 (22,591 ha deforested), which was equivalent to 10.3% of the deforested area in the country [62]. The main deforestation driver is the conversion of forest to pastures for cattle ranching and agriculture. At a lower intensity, unlicensed logging, crop production and mining also contribute [56,61,63–65]. Forest fires may occur during the conversion of forest to facilitate logging [61,66]. For 2018, this region reported 17,719 ha of burned areas, 8266 ha of which were identified as forests. In Manantlán, we used 56 cloud-free images covering 93,153 ha and in Cartagena del Chairá, we used 87 images covering 31,922 ha.

### 2.2.2. Change Detection Algorithm

A wide range of machine learning algorithms is currently available. These algorithms have different strengths and weaknesses, but the success of their application relies on the relevant specific circumstances and data characteristics [67–69]. Nonlinear models have emerged as effective solutions for many real-world classification problems, such as nonlinear support vector machines (nl-SVM) and ensemble methods, such as random forest (RF) and gradient boosting (GB).

Amongst these, RF [70] was deemed the most suitable method due to it providing the best compromise between model accuracy and processing speed. Both SVM and gradient boosting (GB) algorithms are close to RF in their overall accuracy, but they take longer to optimize the parameters via the grid search cross-validation due to limited parallelization. We selected RF due to its easy parallelization and computational speed in comparison to SVM. Deep neural networks (DNN), which have gained increasing attention in the recent machine learning literature, may offer a slight increase in accuracy. However, the computational demand for the training and model calibration using DNN is significantly higher than the RF and, thus, is impractical for a model that has to be regularly updated. Still, our system was built to be flexible and could be adapted to other any classifier (i.e., SVM, DNN) based on specific research needs.

RF is an ensemble classifier, where a group of typically deep CART classifier probabilities are averaged to predict the most likely class label of a pixel or object [70]. Each tree within the ensemble employs random feature selection and bootstrap sampling with a replacement for every split of the CART tree, followed by the averaging of the probabilities across all trees, which aims to mitigate against high variance [70]. For a more detailed explanation of the underlying mathematics of RF, see [70] and [71]. Random forests also offer some interpretability, such as the "out of bag error" and "feature importance" measures, allowing some insight into the inner workings of a particular model [70].

A general rule of thumb with ensemble methods is the more trees, the more accurate the model; although, the error rate may eventually increase beyond a certain number of trees. Hence, for an increasing number of decision trees, there is normally a point of diminishing return that can be approximated to minimize computation time, which is an important factor in a near real-time system. In order to reduce the number of model fits in the subsequent grid search cross-validation, the number of trees was first tested using the out of bag error rate (OOB). This allowed for the fixing of the number of decision trees prior to further hyper-parameter tuning [72].

We calibrated the RF models using training samples from the different land cover change classes (see Table 2) using an exhaustive grid search method where the data were subset into five "folds", with a different fold retained during each iteration to test the model's accuracy and the remainder used to train the model. The contents of each fold were stratified to ensure that they each contained the same percentage of classes as the entire training set. Every possible input parameter combination within the following range was run for each test fold, with the number of decision trees fixed at 500 and 300 for Colombia and Mexico, respectively. The parameter combination that was the most accurate on average across the folds was chosen for scene classification. The models were optimized

using the average accuracy score across the folds. As with scene classification, the change detection used a random forest classier that was trained using the exhaustive methods described above. These were generated using Geospatial Learn [73], which itself in turn uses Scikit-Learn methods [74] applied to geospatial data.

**Table 2.** The training and validation samples per class in each window area (Colombia and Mexico). The training points are pixel-based, while the validation areas correspond to polygons drawn by visual interpretation using PlanetScope data.

| | Classes | Training Sampling Points | | Validation Polygons | |
|---|---|---|---|---|---|
| | | Colombia | Mexico | Colombia | Mexico |
| 1 | Stable Forest | 74,486 | 87,527 | 181 | 123 |
| 2 | Forest > Non-vegetation | 3839 | 8033 | 100 | 50 |
| 3 | Stable Other Vegetation | 6933 | 25,214 | 30 | 20 |
| 4 | Other Vegetation > Non-vegetation | 1439 | 14,455 | 40 | 25 |
| 5 | Stable Non-vegetation | 8294 | 22,039 | 45 | 13 |
| 6 | Non-vegetation > Other Vegetation | 730 | 11,047 | 40 | 8 |
| | TOTAL | 99,047 | 168,315 | 446 | 239 |

### 2.2.3. Targeted Change Classes and Training Data Collecting Protocol

As a supervised classification method, the RF machine-learning model requires a large training dataset for each defined class. For our test areas, we selected six generic categories representing forest-related land cover change based on Table 1. However, from the nine classes in this application of the NRT change detection system, we discarded the classes "Forest > Other Vegetation", "Other Vegetation > Forest" and "Non-vegetation > Forest" because we did not find any examples of these changes. That was expected, considering that this is study used a short time series and these changes occur over a longer period.

Training samples consisted of polygons created by visually digitizing the different classes of study (see Table 1). The samples were collected by examining bi-temporal image stacks for the entire archive of Sentinel-2 data. The training polygons were collected from five to six image pairs over a year to train a more generic model for the different seasons for each window area. Once a sufficient coverage across the AOI and TOI (time of interest) was produced, the polygons/samples were split into training and calibration data using the stratified random split approach. Each pixel within the extent of the training polygons was used as one training data point. A forest baseline map was used to constrain the "Forest > Non-vegetation" class to previously forested areas. The forest baseline map was generated by applying the trained model to the 2017 time series data. A pixel that was classified as forest for most image acquisitions in 2017 and was not deforested within the last 3 months of 2017, was identified as "Forest".

### 2.2.4. Validation Strategy and Accuracy Assessment

A stratified random sampling strategy was employed for validation [75,76]. We validated the total change over the year of analysis so as not to include seasonal variations. Therefore, training samples for each class were generated based on two cloud-free composites that were one year apart. Validation points for each stratum were generated randomly in proportion to the area of that class, with an emphasis on the change classes. We ensured that the standard error of the estimated overall accuracy was below 2% and that, for each stratum, the expected user accuracy was over 70% for the change classes and over 90% for the stable classes (Appendix A, Table A2). With this desired standard deviation for overall accuracy and user accuracy, the proposed validation sampling design is given in Table 2. A Python library was used to automatically generate the validation points based on this approach [43].

Cloud-free composites were generated for a historical baseline mapping period and the monitoring periods. Change detection was applied to the cloud-free composites and then validated. We also compared the NRT change detection accuracies to the monthly products generated from cumulative change detection with a fixed baseline (see training points in Table 2) [20]. PlanetScope image pairs covering the same time period were used for validation [77]. The individual CubeSats of the PlanetScope constellation images had four spectral bands (red, green, blue and near-infrared) with a spectral bandwidth of 60–90 nm and a spatial resolution of 3–4 m. Validation was conducted by overlaying the validation points with the PlanetScope images, visually interpreting the very high-resolution PlanetScope images and assigning the identified change class.

The user, producer and overall accuracies were derived from the confusion matrix to assess the accuracy of the resulting change maps. The user accuracy (Equation (1)) represented the probability that a pixel classified as a given category was a true representation of that category when compared to the validation data. The producer accuracy (Equation (2)) represented how well the reference pixels for the given class were assigned to pixels on the map, i.e., the number of real changes that were correctly detected by the algorithm (see Table 2). The overall accuracy was the percentage of correctly classified pixels across all classes.

$$U_i = \frac{p_{ii}}{P_{i.}} \tag{1}$$

$$P_i = \frac{p_{jj}}{P_{.j}} \tag{2}$$

$$O = \sum_{j=1}^{q} p_{jj} \tag{3}$$

where $U_i$ and $P_i$ represent the user and producer accuracy of class $i$ and $O$ is the overall accuracy among all targeted classes. $p_{ii}$ is the number of validation points that are in class $p_{i.}$ on the classified map to calculate the omission error. $p_{jj}$ is the validation points of the reference class $p_j$ in the validation dataset used to calculate the commission error [75,76].

### 3. Results

*3.1. Forest Change Alerts*

From the historical observations for 2018 in the study area, the change detection identified 1,200,000 deforestation pixels in Manantlán and 50,000 in Cartagena del Chairá. In the Manantlán window area, deforestation events mainly happened in the dry season between December and March (Figure 3, Mexico).

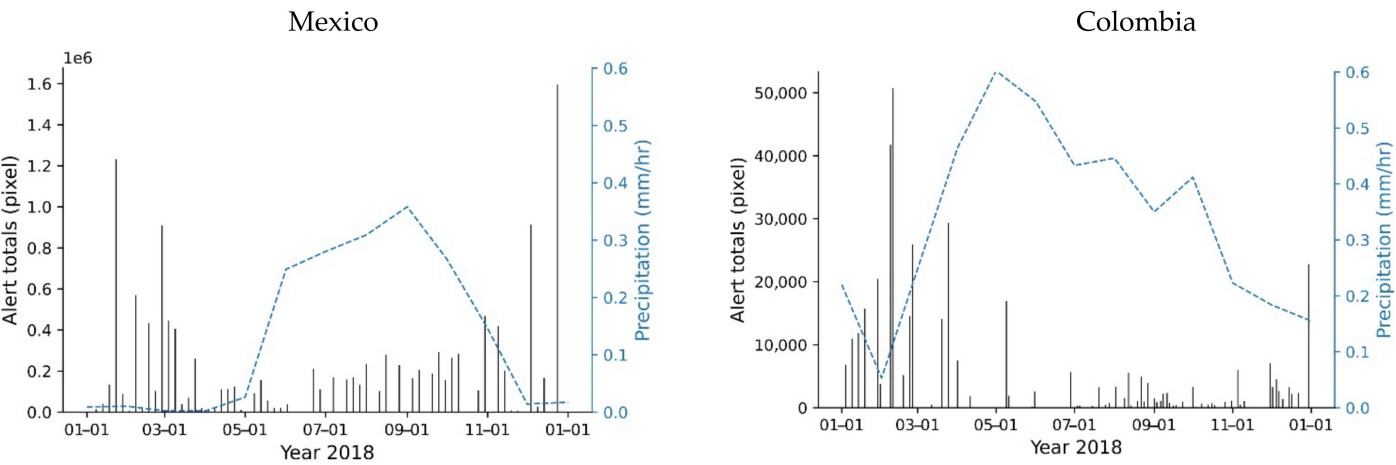

**Figure 3.** The temporal variation in the deforestation alerts for 2018 using Sentinel-2 images in the window areas for Manantlán, Mexico, and Cartagena del Chairá, Colombia.

In total, 0.4% of the forested area in 2017 was disturbed in 2018, with 91% of the detected changes being below 0.04 ha in size and 0.9% of deforested areas being covered by vegetation. A similar trend was observed in the Cartagena del Chairá window area, where deforestation events mainly happened in the dry season between January and March (Figure 3, Colombia). The forest disturbance rate was higher in Cartagena del Chairá than in Manantlán, with 2.2% of forested area being disturbed in 2018. The Colombian study area was smaller than the Mexican area. The size distribution of deforestation events in Cartagena del Chairá showed more smaller disturbances than in Manantlán, where deforested areas were larger in size (Figure 4). In total, 90% of the detected changes were below 0.18 ha and 0.9% of the deforested areas were covered by vegetation.

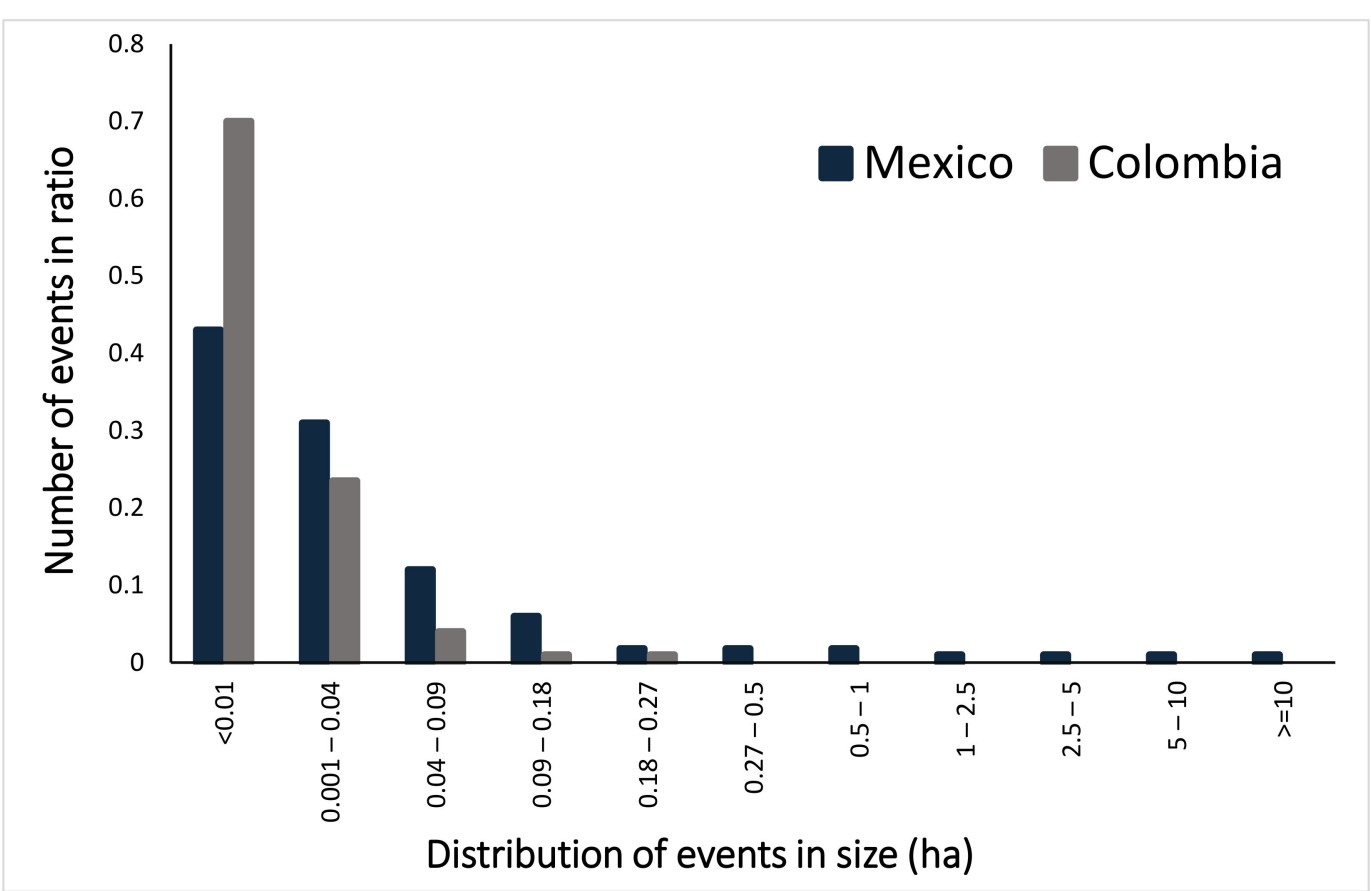

**Figure 4.** The proportional size distribution of the detected deforestation events for the Colombia and Mexico window areas.

Even though this NRT system recorded change alerts as and when a new image was acquired, in Figure 5 we present the total change for all classes for 2018 in both window areas. In Mexico, most changes occurred near settlements and areas of active land use. We observed changes in regrowth and the loss of vegetation other than forest, as well as deforestation in areas of pristine forest (zoom window in Figure 5A). In Colombia, the study area was not close to big settlements. Therefore, the forest loss occurred in the natural Amazonian forest and the changes in other vegetation classes followed land management practices, such as crop growth and harvesting (Figure 5B).

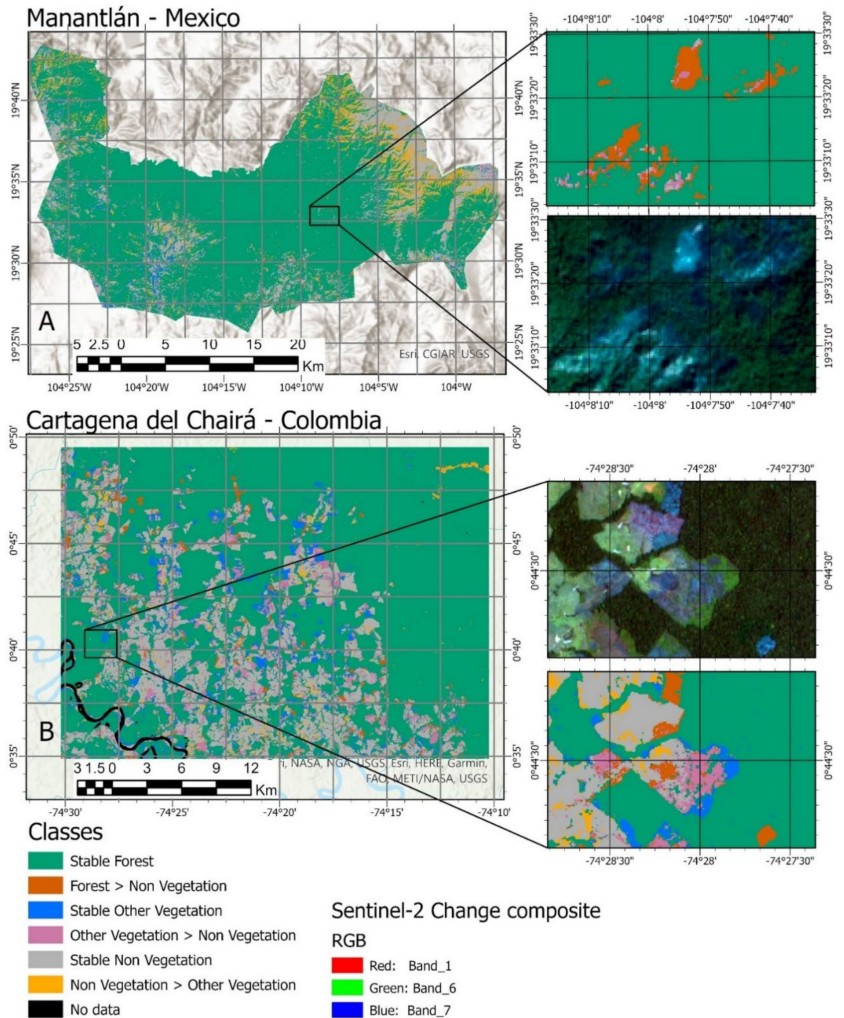

**Figure 5.** The detected forest change events for the window areas in 2018. (**A**) Mexico. (**B**) Colombia. In the RGB Sentinel-2 change composite, red corresponds to the blue channel of the earliest date, green corresponds to the green channel of the latest data and blue corresponds to the red channel of the latest acquisition date. In the RGB composite shown in the zoom windows, areas of dark blue correspond to forest loss, while the purple areas correspond to changes from other vegetation to non-vegetation.

### 3.2. Change Detection Accuracy

The results from the random forest algorithm with the trained forest cover change model achieved an overall accuracy of 92.5% for the automated deforestation alerts from Sentinel-2. Table 3 shows the estimated user, producer and overall accuracies for each change class. The accuracies were sufficiently high to make the PYEO forest cover change detection system useful for local, regional or national forest monitoring for measurement, reporting and verification (MRV) purposes under the REDD+ initiative (Reducing Emissions from Deforestation and Forest Degradation in Developing Countries). The system showed higher accuracies for detecting stable land cover classes and sharp changes, such as forest loss, while there were lower accuracies (65–80%) in the detection of the changes in other vegetation. The lowest accuracy (42%) was observed in the class of non-vegetated areas changing to other vegetation in the Manantlán window area, mainly in the steep mountains. We also found that the omission/commission errors corresponded to broader classes relating to other vegetation and non-vegetated areas. These errors were related to brighter parts of the images due to atmospheric haze.

**Table 3.** The accuracy assessment of the change detections from Sentinel-2 by the visual interpretation of PlanetScope data for the two window areas. NRT: near real-time; PA: producer accuracy; UA: user accuracy; OA: overall accuracy.

|   | Classes | PA (%) | | UA (%) | | OA (%) | |
|---|---------|--------|--------|--------|--------|--------|--------|
|   |         | Colombia | Mexico | Colombia | Mexico | Colombia | Mexico |
| 1 | Stable Forest | 94 | 95 | 96 | 100 | 95 | 97 |
| 2 | Forest > Non-vegetation | 75 | 88 | 99 | 97 | 87 | 93 |
| 3 | Stable Other Vegetation | 90 | 95 | 68 | 67 | 79 | 81 |
| 4 | Other Vegetation > Non-vegetation | 95 | 74 | 59 | 77 | 77 | 76 |
| 5 | Stable Non-vegetation | 93 | 86 | 88 | 40 | 90 | 63 |
| 6 | Non-vegetation > Other Vegetation | 75 | 42 | 100 | 100 | 88 | 71 |
|   | TOTAL | 87.0 | 87 | 80 | 85 | 80 | 86 |

## 4. Discussion

The main advantages of the PYEO forest alerts system using Sentinel-2 data over other forest monitoring systems are the timeliness and speed of detection being within hours after a new image becomes available. The system was designed with the speed of detection in mind so that forest managers are able to intervene in forest cover loss events much more quickly than before. Sentinel-2 was chosen as the satellite for the change detection because it provides a new image every 5 days over all global land areas, which gives it an advantage over Landsat 8 and 9. The Sentinel-2 data are also open-access. In addition, Sentinel-2 has a 10 m resolution in the red, green, blue and NIR bands, which means that the images show small-scale forest loss events, down to the identification of individual large tropical trees being logged when their crown cover is larger than a Sentinel-2 pixel, i.e., 100 m$^2$. The PYEO software package can be adapted to be applied regionally or even globally, although it was not created to generate global forest cover change maps. Instead, it was designed with the user in mind. Hence, the software allows a bottom up, regional approach to change detection based on locally trained machine learning models with user-specified classes of change between land cover types. This gives the user greater control over the definitions of change classes of interest and makes the system very flexible and adaptive. For example, it can be adapted to national or regional forest definitions. This change detection system was based on forest cover loss, as well as the change detection of other change classes that are useful for identifying the transition between other land cover classes.

In this paper, we tested the PYEO NRT forest alert system in two different ecoregions of the tropics: Manantlán in Mexico and Cartagena del Chairá in Colombia. Both areas have been identified as having high deforestation rates over the past years. Nonetheless, they have different ecosystem conditions, as well as different deforestation drivers and patterns. For instance, Manantlán is in a cloud forest in a mountainous region, where the biggest contributions to deforestation are forest fires, agriculture practices, livestock grazing, pine timber extraction and gold mining. In addition, the area presents forest degradation due to illegal logging and gold mining [52,53]. On the other hand, Cartagena del Chairá is on an alluvial plain, where the conversion of forest land to pastures for cattle ranching, as well as illicit agriculture and mining, is rapidly and indiscriminately changing the ecosystems [1,2]. Nonetheless, both areas presented high accuracies for the detected land cover change classes, and particularly high accuracies (>87%) for detected deforestation, for which the model was trained.

Producing a time series change detection analysis using optical data will always make it difficult to maintain the frequency of the detections of the stable classes, particularly in tropical forest. In our study regions, we observed that Manantlán was more affected by clouds on a yearly basis since 11% of the area was permanently covered by clouds, while

only 0.65% of the area at Cartagena del Chairá could not be observed during the period of study.

We compared the PYEO forest loss alerts in 2018 to the Global Forest Watch (GFW) tree cover loss data [9], which is the most established operational global forest monitoring system. There are marked differences between both systems. Firstly, the GFW was designed to detect abrupt changes in forested pixels (deforestation), whilst the PYEO system classifies change between land cover classes directly, not only forest gain and loss but other types of change as well. Secondly, the GFW uses 30 m resolution Landsat images whilst PYEO is based on 10 m Sentinel-2 data. Therefore, it is able to identify a much greater number of small areas of probable forest cover loss, as observed in Figure 6.

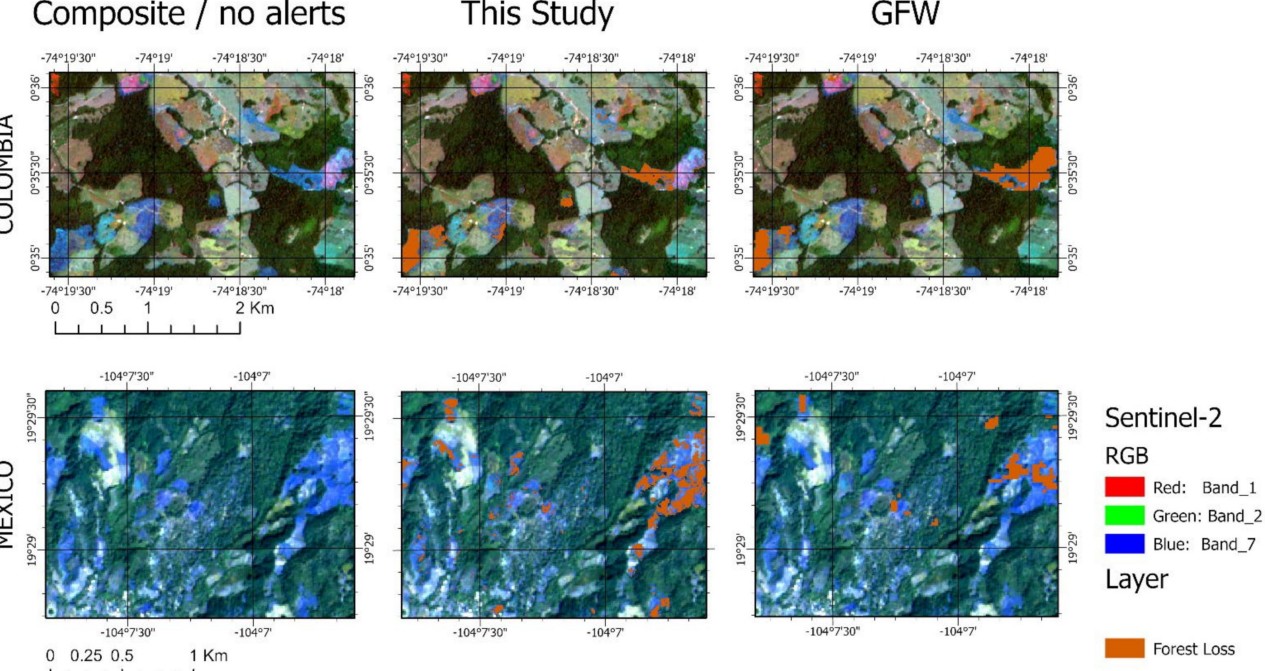

**Figure 6.** A comparison of PYEO deforestation alerts to Global Forest Watch over two small areas in Colombia and Mexico, showing a color composite of Sentinel-2 data used for the automated change detection in this example from January to December 2018. The color composite has the combination of the earliest acquisition date and the date of change detection. In the Sentinel-2 RGB composite image, red corresponds to the blue channel of the earliest date, green corresponds to the green channel of the latest date and blue corresponds to the red channel of the latest date. On the maps to the left, dark blue corresponds to forest cover loss, while in the center and to the right, forest cover loss detections are overlaid in orange. GFW: Global Forest Watch.

While it is possible that some of the very small forest cover loss detections could be false alarms, this comparison demonstrates the added information content in the higher spatial resolution images of 10 m instead of 30 m. In addition, the finer spatial resolution from Sentinel-2 enables a better detection of forest degradation, which can be defined as a "state of anthropogenically induced arrested succession, where ecological processes that underlie forest dynamics are diminished or severely constrained" [74]. While forest cover loss data do not allow for any direct inferences about anthropogenic or natural causes, they do enable the detection of areas in which ecological processes in a forest are beginning to be disturbed. GFW was designed to automatically identify changes based on spectral differences from a global training sample database [16]. PYEO was designed to be trained on local conditions and, although it can be used globally in principle, it was not designed to produce global maps.

Comparing the forest loss reported by the GFW system and the PYEO alert system for 2017, the GFW reported 67,373 deforested pixels (6635 ha) for the Colombia window area

whilst PYEO resulted in 156,629 smaller pixels (1566 ha). Hence, GFW reported nearly four times more deforested area than our system. For Mexico, the PYEO alert system identified 29,748 pixels (292.48 ha), about 13 ha more than the 3097 pixels from GFW (278.73 ha). This difference could be associated with the spatial resolution and the fact that we included the class "Other Vegetation", which could be identified by GFW as forest. In Colombia, we identified 437,147 pixels (4371 ha) classified as "Other Vegetation > Non-vegetation". For Mexico, the class "Other Vegetation > Non-vegetation" was almost non-existent whilst for Colombia, that class covered a large area (4371 ha), as seen in Figure 5.

In developing, testing and applying the PYEO forest alert system, we identified some current limitations of its application. Firstly, seasonally dry tropical forest poses a challenge to training a machine learning model from Sentinel-2 image pairs due to the strong influence of the vegetation seasonality on the spectral reflectance. Secondly, the spectral reflectance in the visible and near-infrared domain is also not very good at distinguishing between forest plantations and natural forest. Thirdly, during the processing of the Sentinel-2 images, due to the fully automated nature of the processing chain, sometimes residual atmospheric effects, such as haze, cloud fringes and cloud shadows, can remain in the images and lead to spurious change detections. The timeliness of the forest alerts depends on the availability of cloud-free pixel observations; hence, the 5-day updates are the best-case scenario that is achievable from free and open data. Finally, to train a good machine learning model requires the representative sampling of the landscape areas, which requires substantial skill and effort on the part of the image interpreter. As a result, the uncertainty of the estimates or false negative detections could be reduced.

## 5. Conclusions

This paper presented a NRT change detection system based on a Sentinel-2 monitoring system. It was developed to detect deforestation events, as well as changes in other land cover classes, at 10 m spatial resolution in a fully automated manner and using a bottom up approach. The system was based on the Python package "PYEO" (see the Supplementary Material for GitHub location). It is flexible and can be adapted to any classifier (i.e., RF, SVM, etc.) based on the user requirements. In addition, it is thought to be useful for all landscapes since it is built on a local training approach, which can be adapted to local landscapes. Nonetheless, local training data need to be collected to replicate the model. We presented the application of the NRT alert system to two window areas in Mexico and Colombia. The results showed that deforestation detections based on this system achieved user accuracies of 97% and 99% for the two window areas, respectively.

This system is highly useful as an automated forest monitoring system, where frequent information is required to improve forest governance. The forest alerts enable the frequent monitoring of deforestation events from space. Hence, it can provide vital information for forest managers and governments regarding the disturbances in their forest land and enable rapid interventions. In this way, the system can contribute to the implementation of the COP26 goal to end deforestation worldwide by 2030.

**Author Contributions:** Conceptualization, C.R., Y.G., V.L., J.F.R., P.R.-V., A.M.P.-P. and H.B.; methodology, Y.G., A.M.P.-P., V.L, P.R.-V., J.F.R. and H.B; formal analysis, Y.G., A.M.P.-P. and V.L.; validation, A.M.P.-P.; original draft preparation, A.M.P.-P., Y.G., H.B. and V.L.; supervision, H.B.; writing—reviewing and editing, A.M.P.-P., Y.G., V.L., P.R.-V., J.F.R., H.B., C.R., P.d.C.B., F.D.B.E.-S., C.U., G.G., E.C., I.P.P.C., M.A.C.S., O.C.N., C.M. and M.I. All authors have read and agreed to the published version of the manuscript.

**Funding:** This research was part of the Forest 2020 project funded by the Global Challenges Research Fund for the UK Space Agency's International Partnership Program, Forest 2020 project, within the frameworks of the Earth and Sea Observation System (EASOS) Malaysia project and the Forests 2020 project. This work was also supported by the Natural Environment Research Council's National Centre for Earth Observation (NCEO).

**Data Availability Statement:** Github location: Pyeo https://github.com/clcr/pyeo (accessed on 1 October 2021).

**Acknowledgments:** We acknowledge to the Global Challenges Research Fund and the UK Space Agency's International Partnership Program to fund the Forests 2020 project. Also supported by EASOS and the National Centre for Earth Observation (NCEO). Forest Sentinel was supported by NERC "REDD+ Monitoring Services with Satellite Earth Observation" (NE/N017021/1). All data were processed on the ALICE high performance-computing cluster managed by the University of Leicester. We thank all partners within the projects for their collaboration, data sharing and infrastructure. We also thank the European Commission Copernicus and Planet Team for the free and open data.

**Conflicts of Interest:** The authors declare no conflict of interest.

## Appendix A

*Appendix A.1. Model Calibration Results*

A fixed number of trees for scene models and change models was established at 500, where OOB errors level out.

Whilst log2 initially appeared to be more accurate, the "number of features" parameter changed in effectiveness during the more rigorous cross-validated grid search. The scene map and change map models consisted of the following hyper-parameters and internal average accuracies in Figure A1 (Appendix A).

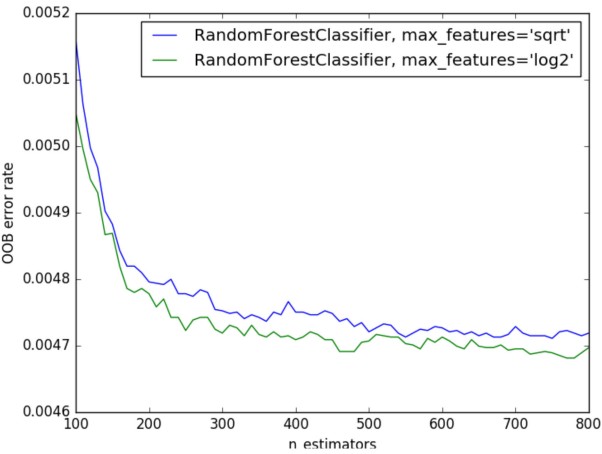

**Figure A1.** The fine tuning of the RF model. OOB is the error rate for change detection measured for the no. of trees (n_estimators), log2 and square root of the number of features choice for each tree node split.

**Table A1.** The cross-validated grid search results for the hyper-parameters of the scene and change models.

| Value | Scene Model | Change Model |
|---|---|---|
| Number of trees | 500 | 500 |
| Split criterion | gini | gini |
| Maximum tree depth | None–increased until pure | 10 |
| Maximum features | square root | square root |
| Minimum leaf samples | 5 | 5 |
| Minimum split samples | 2 | 2 |

It is notable that the optimal parameters are only similar to the tree depth, differing between scene and change models. The feature importance is plotted for each of the image bands used to build the scene and RF change models, which are based on the mean information gain (gini impurity or entropy) per split across the trees. The most important

features in the segregating vegetation and non-vegetation classes that were analyzed in this study are blue, green, red edge, third near infra-red (NIR3) and short-wave infra-red bands (Figure A2 Appendix A).

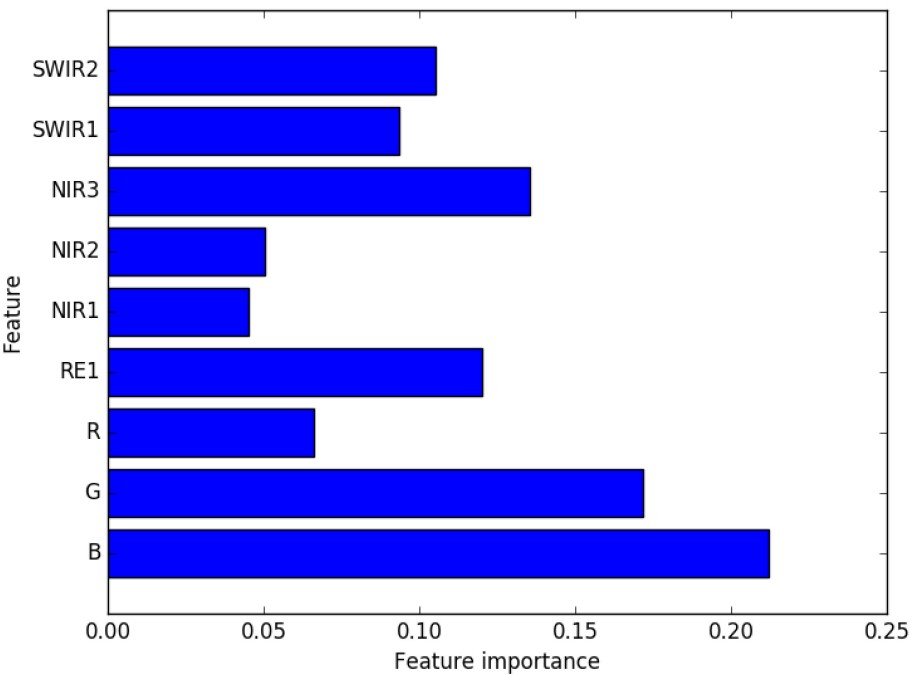

**Figure A2.** The feature importance calculated from the RF scene model. The numerical designations per band indicate where a wavelength area (e.g., red or NIR) is subdivided into smaller bands.

For the change model, the "before" green, "after" green and red band features have the highest importance values, suggesting that they have the most discriminative power for the classes in the change model (Figure A3).

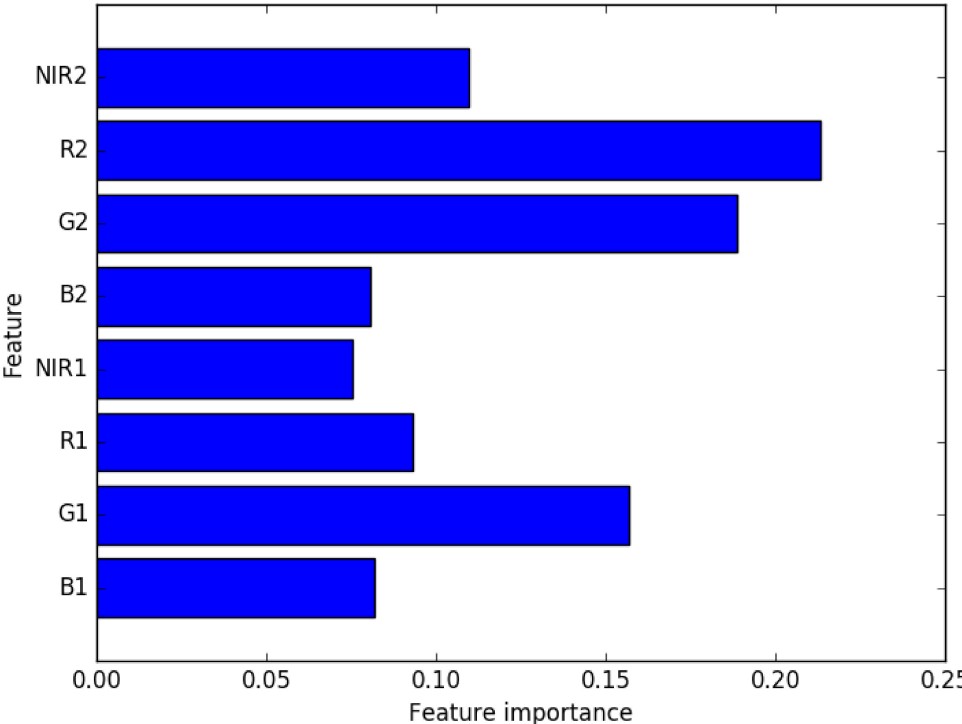

**Figure A3.** The feature importance calculated from the RF change model.

Such changes are in line with the expectations of the input change imagery, where there will be data groupings associated with the changes in the green values from both time slices as vegetation is removed. The feature importance of red band 2 may be due to the exposure of bare soil cover in the "after image", represented as a sharp increase in red values. The remaining features are of similar importance, though the NIR band 2 feature is of greater importance than the remaining bands.

*Appendix A.2. Validation*

**Table A2.** The expected user accuracy.

|   | Classes | Expected User's Accuracy |
|---|---------|--------------------------|
| 1 | Stable Forest | 0.9 |
| 2 | Forest > Other Vegetation | 0.7 |
| 3 | Forest > Non-vegetation | 0.7 |
| 4 | Stable Other Vegetation | 0.8 |
| 5 | Other Vegetation > Non-vegetation | 0.7 |
| 6 | Non-vegetation > Forest | 0.7 |
| 7 | Stable Non-vegetation | 0.9 |
| 8 | Non-vegetation > Other Vegetation | 0.7 |
| 9 | Forest > Other Vegetation | 0.7 |
| 10 | Water Bodies | 0.9 |

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
