# Peer review of "Near Real-Time Change Detection System Using Sentinel-2 and Machine Learning: A Test for Mexican and Colombian Forests"

_remotesensing, doi:10.3390/rs14030707_

Round 1
Reviewer 1 Report
Some minor comments to a fair good paper:
1.- In Figure 1 symbols in the legend are displaced. Moreover, you mention "S2" but it does not appear in the figure.
2.- Perhaps for the reader it may be useful to estimate the area affected by clouds in a yearly basis. Are there any areas where there may be permanent cloud cover problems or where cloud cover may happen in a season relevant for detection?
3.- In the comparison with GFW (specially in the Colombian case) it will be useful to know the share of your classes (table 1) in this area of discrepancy
4.- line 516 "the class other vegetation to non-non veg-etation was almost inexistent" may be "the class other vegetation to non vegetation was almost inexistent"
5.- are there forest fires in the study areas? How do you take this into account? If this factor is not relevant please state it.
6.- are there windthrown event in the forest of the study areas? How do you take this into account? If this factor is not relevant please state it.
Author Response
Dear Reviewer
Thank you very much for your time to comment through this manuscript, your comments and suggestions were appreciated to improve the manuscript. We have reviewed and modified the document accordingly. In addition, the manuscript was thoroughly proof-read. Please see below our responses to your specific comments and suggestions.
- Figure 1 symbols in the legend are displaced. Moreover, you mention "S2" but it does not appear in the figure.
We have revised figure 1 accordingly. Thank you for your suggestion. See line 183.
- Perhaps for the reader it may be useful to estimate the area affected by clouds in a yearly basis. Are there any areas where there may be permanent cloud cover problems or where cloud cover may happen in a season relevant for detection?
Thank you for your suggestion. I have added a paragraph in the discussion providing this information, in line 506-508.
- In the comparison with GFW (especially in the Colombian case) it will be useful to know the share of your classes (table 1) in this area of discrepancy
I have explained later in the paragraph that the discrepancy might be related to the 'other vegetation class' (line 532-541). To illustrate it better, I included the area of change of other vegetation to non-vegetation (line 540).
- line 516 "the class other vegetation to non-non vegetation was almost inexistent" may be "the class other vegetation to non vegetation was almost inexistent"
Thank you for showing this typo. It was corrected
- Are there forest fires in the study areas? How do you take this into account? If this factor is not relevant please state it.
Both areas were affected by forests fires, in Manantlán more than in Cartagena del Chairá. Additional information on forest fires in the study regions was added to section 2.2.1 in lines 261-262 and 281-283. However, consider that we do not specify the causes of detected deforestation events in our algorithm. Therefore, we did not consider vegetation fires.
- Are there windthrown event in the forest of the study areas? How do you take this into account? If this factor is not relevant please state
After consulting the local experts (also co-authors of this paper), we confirm that the study areas were not affected by windthrown events as a main deforestation driver.
- English Proofreading
The manuscript was thoroughly proof-read.
The document submitted is in track changes as requested by the journal. Please consider when reviewing the lines above mentioned, that the number of the lines will match if selecting “hiding the track changes”.

Reviewer 2 Report
The paper is very interesting, original and deals with very current issues.In my opinion, an editing of the English language must necessarily be done. The abstract is too succinct and schematic and should be improved. After the minor revisions, the paper can be accepted.
Author Response
Dear Reviewer
Thank you very much for your time to comment through this manuscript, your comments and suggestions are very much appreciated to improve the manuscript. We have reviewed and modified the document accordingly.
We improved the abstract and proof read the document. Please see attached the manuscript.

Reviewer 3 Report
Brief summary
This work presents a “near-real-time change detection system of disturbance over forests” which automates data downloading, processing, analysis, validation and alert reporting. Detection of changes is conducted with the use of a machine-learning algorithm which uses a chronologically stacked pair of Sentinel-2 images. The main advantage of the system is its timeliness and detection speed (declared as “hours after the new image becomes available”).
The research does not bring any novelty to LULC change detection with satellite data, but presents workable and tested system which allows very detailed and frequent forest cover change detection.
The article is well structured, contains appropriate analyses. Some aspects should be better described or extended in order to guarantee a better understanding of authors' approach. Several problems/doubts were identified and summarized in the following general comments and in the specific comments section.
General comments
I) The change detection approach, used in the presented system, is based on detection of gain and loss in “forest” class and other LCLU classes. However this part of the used methodology was not described broadly enough in the manuscript.
II) The distinct feature of the presented system is near-real-time processing/analyse (change detection). Because of this the authors justify the choice of Random Forest (RF) classifier among others. The Sentinel-2 images with acquisition frequency of 5 days was chosen as data source. An important question related methodology arises: are the differences in processing speed between RF classifier and other classifiers so big, taking under consideration that every next images for an analysed AOI will be available in 5 days, to choose RF as most appropriate classifier. If the required processing time for all of mentioned classifiers equal several or several dozen hours or even 2-4 days, the main important reason to choose appropriate classifier is accuracy. Additionally any artificial neural network architecture, including deep neural networks, realises parallel processing.
Any way, if processing time of taken under consideration classifiers is less than 5 days and their accuracy is similar (and accepted), the authors may choose any of them based on other reasons important for the system/solution preparation.
III) There is a lack of precise information relating how the model “have to be regularly updated.” (ln 281). Does it mean that model update have to be prepared for every other AOI, or when new Sentinel-2 images (for the same AOI) are available i.e. every 5 days?
Specific comments
-
lns 43-44: The sentence is unclear.
-
Figure 1: A) “S2” wasn't used in the diagram. B) How 2 images (at the bottom of the diagram) are connected into the diagram? C) The “composite” and “composite images” had not been clarified prior to the presentation of the Figure 1. This causes confusion during the figure riding – clarification/explanation is needed.
-
lns 181-210: What the (i)-(vii) points describe?
-
lns 183-185: The sentence requires english grammar improvement.
-
lns 194-195: This is unclear and requires improvement.
-
ln 280: I suggest to change “previous solutions” to “RF”.
-
lns 283-285: These sentences require improvement.
-
ln 287: The “the error rate may eventually increase beyond a certain number of trees.” in unclear.
-
ln 317: How were these samples/polygons selected, randomly or by some rule?
-
lns 320, 330, 438: What does “window area” mean? If it the same as AOI, it better to use AOI?
-
ln 339: The literature under nr 20 dosn't provide information on “stratified random sampling strategy”.
-
ln 340: What is “NRT image pair”?
-
lns 359-360: The sentence is unclear.
-
lns 371-376: The “pij” isn't used in equations (1)-(3), and “pii” and “pjj” is used but is not described in the text.
-
ln 419: What is “PYEO forest cover change detection system”? The authors present “NRT change detection system” in the manuscript.
-
Figure 5: A) The legend and the scale bar must be corrected. B) It is mistake to place the north arrow on such maps. C) What “RGB” part of the legend explains? The colors shown in the legend do not appear on the map.
-
lns 443, 464, 490, 520, 532: “PYEO forest alert” or “PYEO” is unclear or inappropriate?
-
ln 452: The “trees being logged if their crown cover is more than about 100 m2” is unclear in the context of the sentence presented in lines 449-452.
-
ln 455: The “locally trained machine learning models” in unclear.
-
lns 458-460: The first part of the sentence is unclear.
-
lns 460-462: This statement provides important premise for a part of used methodology (described in lines 458-460), but it isn't supported with the results of this reserach or with a citation of works on this subject.
-
ln 477: The “PyEO forest loss alerts product” is unclear.
-
lns 477-478: From what year GFW data was used in the comparison? The publication [9] comes from 2013.
-
lns 482-483: The sentence is unclear.
-
Figure 6: A) Looking at the description on the left, the figure presents not only the Mexico area – therefore improvement of the title of the figure is suggested. B) What does the leftmost picture show, there is no description? B) The legend and the scale bar must be corrected. C) It is mistake to place the north arrow on such maps. D) Are “Color-composite” and “RGB composite” have the same meaning?
-
ln 504: What is “FGW”? If it is a typo and it must be “GFW”, why is [74] cited instead of [9].
-
ln 530: The sentence is unclear and need to be corrected.
-
Literature: Description of some literature positions is not complete, e.g.: 10, 40, 49, 69, 71.
-
lns 720-721: The title of the figure is unclear. What is “scene models” and “change models”? What is “log2” and “square root of features”?
-
lns 724-726: The sentence is unclear.
-
ln 726: There is no Figure 8 in the text.
-
Appendix 1, Figure 2 and 3: The titles in unclear. What does "Numerical designations per band indicate before (1) and after (2) respectively" mean?
-
ln 736, 744: The explanation of what the "the most discriminative power in identifying" means is needed.
-
ln 737: The "and 3" is unnecessary here.
-
ln 745: There is no Fig. 13 in the text.
Author Response
Dear Reviewer
Thank you very much for your time to comment through this manuscript. Your comments and suggestions are very much appreciated to improve the manuscript. We have reviewed and modified the document accordingly.
Please find in the table below a document relating the responses to your specific comments, questions and suggestions, and attached the manuscript with the corrections.
Review |
Response |
|
1 |
The abstract is too succinct and schematic and should be improved. |
Thank you for your suggestion. The abstract was improved accordingly. |
2 |
The change detection approach, used in the presented system, is based on detection of gain and loss in “forest” class and other LCLU classes. However this part of the used methodology was not described broadly enough in the manuscript. |
Thank you for your comments. Our approach is based on a machine learning model trained with changes directly, instead of a simple gain and loss in ‘forest’ class between two observation dates. We have added further explanations to clarify our change detection approach in sections 2.2.2, 2.2.3 and 2.2.4. |
3 |
The distinct feature of the presented system is near-real-time processing/analyse (change detection). Because of this the authors justify the choice of Random Forest (RF) classifier among others. The Sentinel-2 images with acquisition frequency of 5 days was chosen as data source. An important question related methodology arises: are the differences in processing speed between RF classifier and other classifiers so big, taking under consideration that every next images for an analysed AOI will be available in 5 days, to choose RF as most appropriate classifier. If the required processing time for all of mentioned classifiers equal several or several dozen hours or even 2-4 days, the main important reason to choose appropriate classifier is accuracy. Additionally any artificial neural network architecture, including deep neural networks, realises parallel processing. |
Thank you for your comments. Model accuracy, along side with training time is a big factor in our decision in choosing the classifiers (explained in line 296). Besides the presented results, we have tested multiple classifiers, included maximum likelihood, SVM and RF. For each classifier we applied fine tuning to ensure the best model was trained. RF and SVM score higher accuracies compared to maximum likelihood. SVM and RF scored similar accuracies but computationally RF was faster than SVM. High accuracy and fast training time combined was the main reason why we selected RF over SVM. This is also reflected in our results with a very high detection accuracy of the deforestation class.
Previous study evaluating different machine learning classifiers (including Neural network) in land cover change applications has also concluded that RF generally performed the best (Fernandez-Delgado et al., 2014).
It would be very interesting to test the performance of Neural network in our study area in the future. However, this paper is aimed at presenting the NRT detecting framework, which can digest any trained machine learning model including those based on NN. . We added information in the paper about this (see lines 301-306)
Hernández-Delgado, M., Cernadas, E., Barro, S., Amorim, D., 2014. Do we need hundreds of classifiers to solve real world classification problems? The journal of machine learning research 15, 3133–3181. |
4 |
Any way, if processing time of taken under consideration classifiers is less than 5 days and their accuracy is similar (and accepted), the authors may choose any of them based on other reasons important for the system/solution preparation. |
Thank you for your suggestion. As we have explained in your previous comment, the system presented was built to be flexible and can be adapted to any type of classifiers. RF was an example in this paper. We have further clarified it in lines 303-308. |
5 |
There is a lack of precise information relating how the model “have to be regularly updated.” (ln 281). Does it mean that model update have to be prepared for every other AOI, or when new Sentinel-2 images (for the same AOI) are available i.e. every 5 days? |
Thank you very much for pointing this out. New models are generally trained over different regions to ensure the model is adapted to local vegetation types and change patterns. If the new AOI has a distinct forest type or change pattern compared to our existing AOIs, then the model need to be updated. Moreover, although we do not update the model for every new Sentinel-2 image, we update the model regularly on an annual basis to achieve better performance as more samples we have over different vegetation types representing a same class (i.e. ‘other vegetation’ bracken vs crops’) as better the accuracy.
We added a paragraph providing further in this matter see lns 219 – 230. |
6 |
lns 43-44: The sentence is unclear. |
We have clarified the sentence (Iine 44-45) |
7 |
Figure 1: A) “S2” wasn't used in the diagram. B) How 2 images (at the bottom of the diagram) are connected into the diagram? C) The “composite” and “composite images” had not been clarified prior to the presentation of the Figure 1. This causes confusion during the figure riding – clarification/explanation is needed. |
Thank you for your comment. We revised figure 1 accordingly to correct the errors and better present the connections of each components in the workflow. We added more explanation relating to the two sub figures (figure 1 b and c) in line 185-186. |
8 |
lns 181-210: What the (i)-(vii) points describe? |
Thank you for your comment. Numbers (i-vii) were aggregated to the diagram to provide a better explanation. See figure 1. |
9 |
lns 183-185: The sentence requires english grammar improvement. |
Thank you for your comment. The whole manuscript has went through proof-reading including line 183-185 (current line 173-176). |
10 |
ln 280: I suggest to change “previous solutions” to “RF”. |
We adopted the suggestion (Line 293). |
11 |
lns 194-195: This is unclear and requires improvement. |
The whole manuscript has went through proof-reading including line 194-195 (current line 196 – 198). |
12 |
lns 283-285: These sentences require improvement. |
Thank you for your comments. We have revised and deleted line 283-285 according to your previous comments (comment 3). We improved the general explanation of the RF model and ensured the sentence is clearly presented with proof-reading (current line 308-315). |
13 |
ln 287: The “the error rate may eventually increase beyond a certain number of trees.” in unclear. |
|
14 |
ln 317: How were these samples/polygons selected, randomly or by some rule? |
The samples/polygons were created by manually digitalising the respective categories from a visual interpretation of all available image pairs of the previous year. Once a sufficient coverage across the AOI and TOI (time of interest) were produced, the polygons/samples were split in training and calibration data using stratified random split approach. (see lns 350 to 357). |
15 |
lns 320, 330, 438: What does “window area” mean? If it the same as AOI, it better to use AOI? |
Thank you for your suggestion. We used window area to represent the two study areas used as example in our study (Manantlán and Cartagena del Chairá). AOI is a more general term used to describe a component in the presented framework. That why we think it might be better to separate those two terms. |
16 |
ln 339: The literature under nr 20 dosn't provide information on “stratified random sampling strategy”. |
Thank you for pointing out the error in the reference. We have updated the reference management software and double checked to manuscript to ensure the references orders are correct. |
17 |
ln 340: What is “NRT image pair”? |
Thank you for your comments. We referred to the two cloud-free composites from Sentinel-2 imagery from one year apart. As it was confusing we revised the sentence accordingly (see lns 371-372). |
18 |
lns 359-360: The sentence is unclear. |
We removed the sentence, because it was not adding anything. |
19 |
lns 371-376: The “pij” isn't used in equations (1)-(3), and “pii” and “pjj” is used but is not described in the text. |
Thank you for your comment. We have corrected the description of equations (1)-(3) accordingly in lines 405 to 409 |
20 |
ln 419: What is “PYEO forest cover change detection system”? The authors present “NRT change detection system” in the manuscript. |
Thank you for your suggestion. We added a sentence and reference in the introduction (line 136) to introduce Pyeo - a Python package on which the presented framework is based. |
21 |
Figure 5: A) The legend and the scale bar must be corrected. B) It is mistake to place the north arrow on such maps. C) What “RGB” part of the legend explains? The colors shown in the legend do not appear on the map. |
We corrected the scale bars and deleted the north arrow. In addition, we clarified in the figure caption what RGB means |
22 |
lns 443, 464, 490, 520, 532: “PYEO forest alert” or “PYEO” is unclear or inappropriate? |
We better introduced Pyeo (Python for Earth Observation), a Python package on which the presented framework is based, earlier in the paper (line 136). Please let us know if including that intro still makes unclear or inappropriate the use of Pyeo |
23 |
ln 452: The “trees being logged if their crown cover is more than about 100 m2” is unclear in the context of the sentence presented in lines 449-452. |
This sentence was revised accordingly (line 479-480) |
24 |
ln 455: The “locally trained machine learning models” in unclear. |
Earlier in the paper, we included a broader explanation to our local approach (see abstract and line 216) |
25 |
lns 458-460: The first part of the sentence is unclear. |
We made this more clear see lns 486-488 |
26 |
lns 460-462: This statement provides important premise for a part of used methodology (described in lines 458-460), but it isn't supported with the results of this reserach or with a citation of works on this subject. |
|
27 |
ln 477: The “PyEO forest loss alerts product” is unclear. |
Thank you for your comment. We have added introduction to Pyeo and its reference earlier in the introduction(line 136) |
28 |
lns 477-478: From what year GFW data was used in the comparison? The publication [9] comes from 2013. |
The comparison was conducted by downloading the annual forest loss product for 2018 covering the same window area from the Global forest watch website. We have cited the recommended paper on the GFW website for this dataset which is reference [9]. |
29 |
lns 482-483: The sentence is unclear. |
We modified the paragraph and corrected this sentence, see ln 511-513 |
30 |
Figure 6: A) Looking at the description on the left, the figure presents not only the Mexico area – therefore improvement of the title of the figure is suggested. B) What does the leftmost picture show, there is no description? B) The legend and the scale bar must be corrected. C) It is mistake to place the north arrow on such maps. D) Are “Color-composite” and “RGB composite” have the same meaning? |
Thank you for your suggestion. We revised figure 6 according to your suggestions |
31 |
ln 504: What is “FGW”? If it is a typo and it must be “GFW”, why is [74] cited instead of [9]. |
Thank you for your suggestion. We have corrected the typo. |
32 |
ln 530: The sentence is unclear and need to be corrected. |
Thank you for your suggestion. We have revised the sentence (line 562) |
33 |
Literature: Description of some literature positions is not complete, e.g.: 10, 40, 49, 69, 71. |
Thank you for pointing out the error in the reference. We have updated the reference management software and double checked to manuscript to ensure the references are correct. |
34 |
lns 720-721: The title of the figure is unclear. What is “scene models” and “change models”? What is “log2” and “square root of features”? |
Thanks for pointing this out. We change the figure tittle to Fin tuning of RF model. log2 and 'square root' are the parameters that you can tune to achieve higher accuracies in a RF model. |
35 |
lns 724-726: The sentence is unclear. |
We improved the sentence. |
36 |
ln 726: There is no Figure 8 in the text. |
Thank you for your suggestion. We have deleted the in-text citation to figure 8. |
37 |
Appendix 1, Figure 2 and 3: The titles in unclear. What does "Numerical designations per band indicate before (1) and after (2) respectively" mean? |
Thank you very much for your suggestion. |
38 |
ln 736, 744: The explanation of what the "the most discriminative power in identifying" means is needed. |
Thank you for your comments. The discriminative power is subset evaluation method used to support the feature importance divisions. |
39 |
ln 737: The "and 3" is unnecessary here. |
We have fixed the issue. |
40 |
ln 745: There is no Fig. 13 in the text. |
We have fixed the issue. Thank you for your comments. |
The corrected manuscript submitted is in track changes as requested by the journal. Please consider when reviewing it, that the number of the lines will mentioned in the responses document will match if selecting “hiding the track changes”.
